# Genomic Landscape of Chromosome X Factor VIII: From Hemophilia A in Males to Risk Variants in Females

**DOI:** 10.3390/genes15121522

**Published:** 2024-11-27

**Authors:** Olivia Morris, Michele Morris, Shawn Jobe, Disha Bhargava, Jena M. Krueger, Sanjana Arora, Jeremy W. Prokop, Cynthia Stenger

**Affiliations:** 1Department of Biology, University of North Alabama, Florence, AL 35632, USA; 2HudsonAlpha Institute for Biotechnology, Huntsville, AL 35806, USA; 3Center for Bleeding and Clotting Disorders, Michigan State University, College of Human Medicine, East Lansing, MI 48824, USA; 4Department of Pediatrics, Michigan State University, College of Human Medicine, East Lansing, MI 48824, USA; 5Department of Neurology, Helen DeVos Children’s Hospital, Corewell Health, Grand Rapids, MI 49503, USA; 6Office of Research, Corewell Health, Grand Rapids, MI 49503, USA; 7Department of Mathematics, University of North Alabama, Florence, AL 35632, USA

**Keywords:** F8 gene, factor VIII, hemophilia A, alternative promoter, bioinformatics, variants of uncertain significance, chromosome X, gene therapy

## Abstract

Background: Variants within factor VIII (F8) are associated with sex-linked hemophilia A and thrombosis, with gene therapy approaches being available for pathogenic variants. Many variants within F8 remain variants of uncertain significance (VUS) or are under-explored as to their connections to phenotypic outcomes. Methods: We assessed data on F8 expression while screening the UniProt, ClinVar, Geno2MP, and gnomAD databases for F8 missense variants; these collectively represent the sequencing of more than a million individuals. Results: For the two F8 isoforms coding for different protein lengths (2351 and 216 amino acids), we observed noncoding variants influencing expression which are also associated with thrombosis risk, with uncertainty as to differences in females and males. Variant analysis identified a severe stratification of potential annotation issues for missense variants in subjects of non-European ancestry, suggesting a need for further defining the genetics of diverse populations. Additionally, few heterozygous female carriers of known pathogenic variants have sufficiently confident phenotyping data, leaving researchers unable to determine subtle, less defined phenotypes. Using structure movement correlations to known pathogenic variants for the VUS, we determined seven clusters of likely pathogenic variants based on screening work. Conclusions: This work highlights the need to define missense variants, especially those for VUS and from subjects of non-European ancestry, as well as the roles of these variants in women’s physiology.

## 1. Introduction

Genetic variants on the human sex chromosomes represent a significant challenge in understanding disease risks for males and females. Males carry a single Chromosome X (ChrX) and a smaller Chromosome Y (ChrY), while females carry two ChrX. Most ChrY elements cannot crossover with ChrX, meaning many variants from a father’s ChrY and ChrX are passed on to 100% of their offspring. This inheritance structure, one not common to other chromosomes of humans, results in the accumulation of phenotypic-associated variants relative to other chromosomes [1], some of which are involved in disease states [2].

Within the early embryonic development of females carrying two ChrX, one of the ChrX copies is inactivated, mainly by the long noncoding *XIST* RNA, through which a random selection of ChrX is inactivated, a process known as XCI [3,4]. This initial XCI progresses through epigenetic changes, resulting in non-random XCI in subsequent cell proliferation events and maintaining the same inactivated ChrX found in the mother cell [5,6]. Thus, clumps of cells in tissues often have the same ChrX inactivated, with variability increasing as distance increases, a common trait which can be observed in the color patches in the calico cat. However, growing evidence is found for skewing, in which ChrX is inactivated, increasing the number of diagnoses linked to ChrX-linked recessive traits, due to alterations in XCI [7,8,9]. Some genes on ChrX escape XCI and have higher expression levels in female tissues than male tissues, driving sex differences [10,11]. ChrX genes vary highly in XCI and escape, depending on an individual’s cell type, age, and other environmental factors [12,13]. Thus, exploring the role of ChrX genetic variants within an individual requires a delicate understanding of sex chromosomes, XCI, XCI escape, and cell type/location of interest.

When a missense variant falls on ChrX within a male and is associated with impacting a protein, 100% of the total protein contains the variant (known as hemizygous) and can result in sex-linked diseases. For example, hemophilia A is caused by genetic variants in factor VIII (F8, OMIM 306700), which are primarily diagnosed in males. In some sporadic cases, homozygous, compound heterozygous (meaning two separate deleterious variants on opposite ChrX), female hemizygous (lacks a second copy of the gene), or heterozygous variants with non-random XCI occur in females with hemophilia A [14,15,16,17,18]. Females with a single heterozygous missense variant with random XCI produce roughly 50% cells with dysfunctional protein and 50% with normal protein. As to the case of a gene such as *F8*, it is uncertain what role these heterozygous variants have in females, but rarely are they associated with hemophilia A. Female carriers of a heterozygous variant in *F8* have a variable phenotype, often considered a non-disease type, in which XCI can influence female phenotypes of bleeding and clotting [19,20]. More work is needed to address the genomic landscape and phenotypic associations for variants in females.

Diagnosing hemophilia A or B has historically been based on biochemical coagulation assays, with recombinant factors used for treatment [21]. There are three different classifications of hemophilia A, based on the activity of the F8 protein. Severe hemophilia A has less than one percent factor activity, moderate hemophilia has one to five percent factor activity, and mild hemophilia has five to forty percent factor activity. Disease severity can be attributed to the variants found in the F8 gene sequence [22].

The replacement therapies for hemophilia A have a short half-life and require continued administration [23]. The treatment costs for a patient with severe hemophilia A can rise above $300,000 annually, with a significant portion of that cost being from the recombinant factors [24]. Gene therapy has progressed for hemophilia A over recent years, with the promise of a treatment with minimal need for re-administration [25]. On 30 June 2023, the FDA approved the factor VIII-encoded AAV5 ROCTAVIAN gene therapy for the treatment of patients with F8 genetic variants (125,720).

If the heterozygous variants within a female result in dysfunctional protein and are passed on to male offspring, they can result in hemophilia A. Thus, carrier screening during pregnancy can be used as an early detection strategy for F8 missense variants. Hemophilia A genomic sequencing and newborn screening have improved diagnosis and outcomes [26,27]. However, recent work in detecting F8 missense variants [28] continues to suggest novel changes, suggesting that sequencing alone, without clotting assays, cannot definitively diagnose hemophilia A.

As sequencing has advanced within populations composed of individuals with bleeding and clotting disorders, so has the identification of variants of uncertain significance (VUS). VUS cannot be reported to patients, and can prevent the proper diagnoses of genetic conditions without clotting assays, especially in female carriers. While management of bleeding and clotting diseases can continue in these individuals if biochemical assays outside of genomics are employed as diagnostics, the lack of genetic diagnoses could prevent the initiation of gene therapy and limit the identification of female carriers and familial screening. In addition, the role of race/ethnicity in F8 variants continues to be challenging in the determination of treatment [29]. This suggests a critical need to refine further the genetic landscape of ChrX genes such as *F8*. This paper addresses the variant landscape of *F8*, utilizing multiple genomic databases; this is followed by a sequence-to-structure-to-function exploration [30] of high-probability functional missense variants.

## 2. Materials and Methods

The UCSC Genome Browser [30] was used to annotate functional genomic data around the *F8* gene on 27 September 2023. The hg19 genome build was used, as more data were mapped to this release. The genome browser was manually positioned to include *F8* and one complete gene flanking on either side (*FUNDC2* and *SMIM9*).

Single-cell expression data for *F8* were extracted from the Human Protein Atlas [31]. Bulk tissue expression was annotated from the GTEx [32] version 8, transcript per million (TPM), mapped to the human Gencode [33] version 26 isoforms. The expression or splicing quantitative trait loci (eQTL and sQTL) data were also extracted for each tissue sample from the GTEx [32] version 8 data, and violin plots and statistics extracted from the GTEx webpage.

Protein sequences were extracted from NCBI Ortholog [34] on 27 December 2023, using the single-protein per-species extraction. Sequences were aligned using MAFFT [35] and manually annotated to remove species with large regions (>10 amino acids) missing. A maximum-likelihood phylogenic tree was generated using MEGA [36] with the Jones–Taylor–Thorton (JTT) model [37] and 50 bootstraps. The alignment matrix was extracted and processed for amino acid conservation, as described in previously published work [38,39], with a score of 0 to 1, where 1 represents 100% conservation. Genomic variants resulting in missense changes were extracted from UniProt [40], ClinVar [41], Geno2MP [42], and gnomAD version 4.1 [43] on 27 December 2023. UniProt was used to extract known posttranslational modification sites (ptms) and enzyme cleavage locations, with a binary score of 1 (is present) or 0 used for each amino acid. ClinVar variants were split into two groups: those annotated as Variants of Uncertain Significance (VUS) and those classified as pathogenic or likely pathogenic. All variants were compiled from the tools, using the amino acid change annotation, and then processed through the variant effect predictor (VEP) [44] with CADD scores [45], SIFT [46], PolyPhen [47], MutationAssessor, ClinPred, DANN, GERP++_RS, GenoCanyon, LIST-S2, MVP, PROVEAN, and REVEL used. A total variant score was generated by adding the conservation, UniProt annotation, and VEP scores for each amino acid change. The population structure and heterozygosity–homozygosity for each variant were extracted from gnomAD. ClinVar and Geno2MP-listed phenotypes were manually curated.

Protein models were generated using YASARA [48] modeling to clean up the protein data bank (PDB) [49] structure file relative to the 7kwo of F8–von Willebrand factor (VWF) interaction. Following energy minimization, the protein model was processed for molecular dynamics simulation in YASARA using the AMBER14 force field [50] in explicit water at 0.997 g/mL (206,641 total water molecules) and a pH of 7.4 for setting pka, with Na and Cl at 0.9% mass fraction (580/577 atoms each) and a periodic cell boundary. Simulation snapshots were collected every 100 picoseconds, with 801 snapshots extending across the 80 nanoseconds of the simulation. Trajectory files were processed with the YASARA md_analyze and md_analyzeres macros. All data relating to the amino acid conservation, variants, and molecular dynamic simulations are available at Appendix A
https://doi.org/10.6084/m9.figshare.26499532.v1.

## 3. Results

### 3.1. F8 Gene Promoters, Isoform Expression, and Noncoding Variants

Understanding *F8* variants first requires an understanding of the *F8* gene expression, the isoforms of the gene, and the role of XCI. *F8* is located on ChrX, between the *FUNDC2* and *SMIM9* genes (Figure 1). *F8* has two major promoter elements, one located near the FUNDC2 gene that codes for the *F8-202* isoform (NP_000123) and another promoter internal to the gene that codes for the *F8-201* isoform (NP_063916). Both promoter elements have CpG islands near them with epigenetic signature features traditionally associated with the promoter sequence (Figure 1, Gene Regulation). The *FUNDC2* gene utilizes a promoter separate from the *F8-202* isoform, one located ~4000 bases apart, which is reflected through the mostly ubiquitous *FUNDC2* expression relative to the narrow *F8-202* expression profile. *F8-202* translation results in a 2351 amino acid protein, while the *F8-201* isoform codes for a 216 amino acid protein. *F8-202* encodes a large glycoprotein which associates with von Willebrand factor in a noncovalent complex. *F8-201* encodes a putative small protein mainly consisting of the phospholipid binding domain of factor VIIIc and is essential for coagulant activity. The exon locations (boxes in Figure 1, Splicing) are enriched for functional human variants (Figure 1, Human Variants) and conservation (Figure 1, Conservation). Multiple commonly inherited variants outside of the exons can be observed that suggest population background-specific linkage disequilibrium (LD) blocks (Figure 1, 1000 genomes).

Based on the Human Protein Atlas single-cell database, *F8* is most highly expressed in endothelial and lymphatic endothelial cells, followed by adipocytes, with lower levels found in other cell types (Figure 2A). Adipose, heart, and breast have the highest tissue expression of *F8*, based on the large human datasets of GTEx (Figure 2B), and reflecting the associated single cell-types shown in Figure 2A. The scope of GTEx covers hundreds of male and female human tissues, allowing us to address potential XCI and XCI escape. Isoform *F8-201* is highly expressed in all tissues of GTEx, relative to the *F8-202* isoform (Figure 2B). Males and females are similar in average expression over all tissues for both isoforms. However, in the breast–mammary tissue, there is a trend towards higher expression of both isoforms in males. Yet, when addressing the expression of both isoforms in various age groups between males and females, no significant differences were observed (Figure 2C). As female samples rarely have double the expression of male samples, this suggests that *F8* XCI is consistent across age groups and tissues. The data show that males tend to have more outliers of both isoforms over various age groups.

To address the role of noncoding variants on *F8* expression levels, we utilized the GTEx expression quantitative trait loci (eQTL) analysis. Nerve–tibial tissue contains the highest number of variants significantly associated with variation in *F8* expression levels, followed by adipose and artery tissues (Figure 2D). In the nerve–tibial eQTLs, the effect size is similar for most variants across ChrX, with some variants being associated with a positive effect-size and some with a negative effect-size (Figure 2E). The *p*-values (as reflected by a −log10) suggest a regional eQTL LD block that is centered around 12 coinherited variants with similar *p*-values and effect sizes (rs5945278, rs4898407, rs5945134, rs73569615, rs115609690, rs114209171, rs7886856, rs17051889, rs1470586, rs73567794, rs5945130, and rs5945131), and covering 50,058 bases (ChrX: 155,001,128-155,051,186 hg38, ChrX:154,229,503-154,279,561 hg19) located between the *FUNDC2* gene and the *F8-202* promoter element (Figure 1, bottom).

Variants associated with a positive effect-size as to *F8* expression are found throughout all populations, based on the gnomAD database, with the highest allele frequencies, 0.3715, being found in South Asian individuals, followed by Admixed Americans (Figure 2F). The variants associated with negative effect-size follow the opposite trend, with the lowest frequency being in the South Asian population background. The LD block for both positive and negative effect-size follows a dosage level of homozygosity and heterozygosity across multiple tissues of GTEx (Figure 2G), suggesting that allele contributions directly influence *F8* gene regulation. As observed in a genome-wide association study(GWAS), variants in the LD block (rs114209171) have been associated with thrombosis and venous thromboembolism [51]. Of these variants in high LD, two fall within known transcriptional regulation sites; this is based on RegulomeDB [52] scores, in which rs1470586 falls within the intron of *F8* of a known active enhancer element of the NHLH1 transcription-factor binding site. NHLH1 has been associated with altered expression in glioblastoma patients with venous thromboembolism [53]. We suggest that future examination of this LD block involved in F8 expression levels and thrombosis risk should involve further evaluation regarding XCI and male hemizygous expression levels, and focus on the sex-stratified GWAS analyses rarely utilized in population genomics.

### 3.2. F8 Protein Conservation and Missense Variant Locations

Hundreds of F8 protein sequences from diverse vertebrate species can be found within the NCBI ortholog database. Extraction of sequences, their alignment, and the removal of those with large deletions resulted in 322 unique species of F8 protein sequence for the longer F8-202 isoform (Figure 3A). These sequences show a high percentage of species with conservation over the A1 and A2 regions of the heavy chain and the A3, C1, and C2 regions of the light chain (Figure 3B). The F8-201 isoform has an alternative first exon not found within the F8-202 isoform, resulting in an alternative first-eight amino acids, followed by the inclusion of amino acids 2144-2351 of the C2 region of the light chain. This protein is known as the F8B protein, and has an unknown function, but appears to be critical for various phenotypes within mice [54]. It should be noted that this alternative N-terminus of the protein in isoform F8-202 results in the extensive loss of the hydrophobic amino acids of the long isoform’s signal peptide, which are required for cell secretion. Thus, it is likely that the smaller protein of the F8-201 isoform is retained in the cell, although there is minimal knowledge of its function.

Missense variants were extracted from five different genomic datasets. The UniProt disease annotated variants and the ClinVar Pathogenic/Likely Pathogenic variants (Figure 3B, red) represent the highly probable impactful changes to the F8 protein which are found at high levels within the A1 and A2 heavy-chain and A3, C1, and C2 light-chain. The ClinVar VUS and Geno2MP variants (Figure 3B, magenta) represent changes within individuals with a noted phenotype but without confident pathogenicity. These variants are found to be evenly distributed throughout F8. The gnomAD dataset (Figure 3B, black at the bottom) represents a population sequencing of variants from 730,947 exome and 76,215 genome sequences, identifying an even distribution throughout F8 but with some elevation found within the non-conserved B region of the heavy chain.

### 3.3. Common gnomAD Variants Connected to Phenotype

Integrating our conservation scores, at each amino acid, relative to the variant list identified a high conservation of UniProt and ClinVar (likely) pathogenic variants. At the same time, the other three databases showed a higher degree of variability of conservation (Figure 4A). Addressing those variants falling on amino acids with >70% conservation, many of them overlap the gnomAD dataset, allowing for the calculation of population-specific allele frequencies (Figure 4B). Many variants found in the disease-associated groups of UniProt and ClinVar are both pathogenic and within the A1 or A2 heavy-chain or A3, C1, or C2 light-chain regions. Those with population allele frequencies > 0.001 and conservation > 70% (Table 1) represent an enrichment within diverse populations, such as Middle Eastern, Admixed American, and South Asian, while demonstrating underrepresentation in populations of European ancestry (Figure 4C). This suggests the far too common issue of genomic-outcome understudying in non-European variants.

A single variant in Table 1, N583S, has a known homozygous female in gnomAD with 17 hemizygous males. In all, 99% of species have a conservation of N at this amino acid, with a single species observed with an S. The variant is annotated in UniProt for a disease, but absent in Geno2MP and ClinVar. It is also enriched in the South Asian population of gnomAD (20 individuals), and almost exclusively observed in male samples (21 males), with 1/5 of females observed at homozygous frequency, suggesting a disbalance in observation.

A total of five variants (Table 1: Q1764R, R458H, P83R, R2016Q, and E2023K) are listed in ClinVar as pathogenic or likely pathogenic and have allele frequencies in diverse populations relative to gnomAD. All five variants lack an observed female homozygous individual within gnomAD at the time of analysis. Q1764R, relatively speaking, maintains amino acid function, with 19 species with R and four species with a K throughout evolution. Q1764R is observed in the gnomAD “remaining” population group with 56 hemizygous male samples. R458H, enriched in the Middle Eastern population, also maintains some amino acid functions, with eight species having an H and 20 males annotated as hemizygous. R2016Q, enriched in African/African American individuals, has 12 species with a Q throughout evolution and three hemizygous males. E2023K, enriched in Admixed Americans, while causing a change alteration in the protein, is found as a K in nine species and is observed in four hemizygous males. The data for Q1764R, R458H, R2016Q, and E2023K suggest that further examination of their annotation in ClinVar may be warranted. P83R, enriched in Admixed American, has a very distinct amino acid; three species were annotated with an R, and three hemizygous male samples were observed, making the data more ambiguous regarding pathogenicity.

Contrasting the ClinVar pathogenic annotations, several of the ClinVar VUS and Geno2MP variants found in diverse populations appear to have more functional consequences. These include E1690G (East Asian), Y657H (Middle Eastern), R612H (Middle Eastern), P1265Q (South Asian), and R2166Q (South Asian). To further process the ClinVar VUS and Geno2MP variants, we utilized an additional 12 functional tools for protein alteration predictions. We summed the predictions relative to the pathogenic and likely pathogenic ClinVar scores (Figure 4D). Only 41 VUS and Geno2MP variants fall within the range of scores associated with high confidence levels among the ClinVar pathogenic variants (Table 2). Of these 41 variants, ten are also found in gnomAD, with the functionally conserved S2125T enriched in the Middle Eastern population, the significant change G525R in South Asian individuals, and the Y1717H also in South Asian individuals.

### 3.4. Exploring VUS with Protein Structure and Molecular Dynamic Simulations

To better define the variants of Table 2, many of which are ultra-rare changes in ClinVar and Geno2MP, we utilized a molecular dynamics simulation approach which has previously been used to establish a movement correlation map for VUS in CFTR [55] and the NMDA receptor complex [56]. The platform enables the interaction of known protein models for F8 with the von Willebrand factor (VWF, Figure 5A) to resolve how each amino acid moves relative to other amino acids. An 80-nanosecond simulation shows the relative stability of amino acid movement over time (Figure 5B), with various amino acids moving at different levels depending on their local environment, which is composed of other residues (Figure 5C). Using a dynamics cross-correlation calculation, the movement of all amino acids relative to each other can be converted into probabilities, identifying amino acids capable of interacting locally and across proteins and domains (Figure 5D). The variants in Table 2 were processed to determine how many ClinVar pathogenic variants correlate in movement within F8 (Figure 5E) or with any amino acids in VWF (Figure 5F). All Table 2 variants are correlated to at least one F8 pathogenic variant > 0.8, while only variants in the light-chain region are correlated with amino acids in VWF.

Viewing the amino acids highly correlated with VUS (Figure 5G, yellow) on the protein structure relative to the known ClinVar pathogenic residues (Figure 5G, side chains shown) and other subunits (Figure 5G, surface plots) reveals seven clusters of variants. The six VUS with the highest correlations to ClinVar pathogenic variants fall in the light-chain C1 + C2 domain. As highlighted below, one of the significant issues with VUS annotations in ClinVar is that they lack reannotation over time and, therefore, could have been reclassified in the interval without this being reflected in a ClinVar reannotation.

**Cluster 1**—Composed of I2100, Y2124, and S2125 of the light-chain C1 + C2 domain contacting the light-chain A3. Isoleucine (ILE, I) at amino acid 2100 has an instance of the unclassified variant (I2100T) and the amino acid has the highest number of highly correlated pathogenic amino acids (Figure 5E), including ILE 2099 (0.984 movement correlation), HIS 2101 (0.984), GLY 2102 (0.972), PRO 2162 (0.969), LEU 2185 (0.969), GLU 2184 (0.958), PHE 2145 (0.947), MET 2183 (0.944), PHE 2120 (0.937), TYR 2124 (0.931), PHE 2146 (0.926), ARG 2182 (0.92), ASP 2093 (0.916), SER 2125 (0.914), ARG 2169 (0.913), PRO 2172 (0.913), ASN 2148 (0.909), THR 2173 (0.909), LYS 2091 (0.908), ARG 2135 (0.905), and SER 2192 (0.9). This amino acid is found as an ILE, VAL, LEU, or PHE throughout evolution, and never as a THR. Conservation values, nearly every functional tool, and its CADD score of 24.4 ranks suggest that this VUS is highly damaging. Like so many variants in Table 2, these metrics suggest the need to reevaluate the pathogenicity of many VUS.

**Cluster 2**—Composed of Y2116, S2082, and A2108 within the internal packing of the light-chain C1 + C2 domain and the interaction with the VWF protein. Alanine (ALA, A), at amino acid 2108, is the third-ranked pathogenic, correlated residue, and is associated with most of the same amino acids found in Cluster 1. The VUS (A2108P) is predicted to be damaging in every tool analyzed, with conservation at 99% as an ALA and only one species demonstrating either a VAL or THR. The PRO of the VUS is likely to disrupt the β-strand of the protein domain, resulting in significant changes to protein structure. Tyrosine (TYR, Y), at amino acid 2116, is the most correlated residue among the VUS (Y2116H) with the VWF protein amino acids LEU 797 (0.945), ASN 794 (0.941), GLU 798 (0.93), GLN 793 (0.924), CYS 799 (0.923), THR 791 (0.922), LYS 790 (0.917), TYR 795 (0.917), ASP 796 (0.916), CYS 792 (0.911), and CYS 788 (0.902). The VUS Y2116H is predicted to be damaging in every tool and is conserved as a TYR or PHE aromatic amino acid in all species.

**Cluster 3**—Composed of Y1717, E1756, and I1756 of the light-chain A3 contacting the light-chain C1 + C2 domain and VWF. Glutamine (GLU, E), at amino acid 1756, is highly correlated with pathogenic changes at PHE 1794 (0.963), GLY 1779 (0.952), ARG 1768 (0.952), LEU 1775 (0.95), GLN 1764 (0.945), PRO 1873 (0.932), GLY 1769 (0.923), LEU 1951 (0.914), and GLU 1770 (0.902). The VUS E1756V is predicted to be damaging in all tools but the GenoCanyon tool, with the evolutionary analysis never observing a hydrophobic amino acid like Val (V). Tyrosine (TYR, Y), at amino acid 1717, is one of the most highly correlated amino acids relative to the VWF protein amino acids LEU 765 (0.925), SER 764 (0.913), and SER 766 (0.902). The VUS Y1717H is predicted to be damaging in all tools but GenoCanyon and is found 100% conserved as a TYR.

**Cluster 4**—Composed of W1854, D1859, and D1861 of the light-chain A3 contacting the heavy-chain A2 domain. Tryptophan (TRP, W), at amino acid 1854, is highly correlated to the pathogenic amino acids ALA 1853 (0.956), PRO 1873 (0.934), SER 1806 (0.927), and MET 1791 (0.916). The VUS W1854C is the second-highest scored VUS in Table 2, with 100% conservation as a TRP and all tools predicting functional outcomes.

**Cluster 5**—Composed of R22, T154, and V159 of the heavy-chain A1 contacting the light-chain C1 + C2. Threonine (THR, T), at amino acid 154, is known to be changed to Alanine (ALA, A) in hereditary factor-VIII deficiency disease and is annotated as likely pathogenic based on a 21 May 2022 submission. The VUS T154I was submitted on 29 January 2020, and thus warrants reannotation. Evolution has eight species with a Valine and three with an Isoleucine at this amino acid. Both overlap the VUS and pathogenic annotations, making this amino acid site important for further wet-lab characterization.

**Cluster 6**—Composed of D169, T174, F214, P309, T314, and A315 within the internal packing of the heavy-chain A1 domain. P309L is the top-ranked variant in Table 2, with all tools predicting pathogenicity and no species within the evolutionary analysis having an LEU. Interestingly, this amino acid has no pathogenic amino acids with correlations above 90%; however, it is located at a critical bend location that the evolutionary analysis maintains with small flexible side chains (SER and ALA), which the LEU does not provide.

**Cluster 7**—Composed of L480, V534, L571, and K575 of the heavy-chain A2, near the heavy-chain A1 contacts. Lysine (LYS, K) at amino acid 575 is correlated with pathogenic variant locations at SER 577 (0.953), CYS 547 (0.953), ARG 546 (0.941), and THR 541 (0.913). The VUS K575T is predicted to be functional in all tools but GenoCanyon and REVEL, and falls on a site conserved as a polar basic residue.

Overall, amino acid analysis combining deep conservation, functional tool predictions, and protein-based molecular dynamic simulations can prioritize variants for further validation.

## 4. Discussion

It is striking that not a single variant in Table 1 is of European ancestry, suggesting a critical need for diverse population analysis of F8 missense changes. F8 is not the only protein with this significant issue; our group has shown similar findings for CFTR [55], which have risen in relevance as triple-therapy treatments for cystic fibrosis have disproportionately failed in non-European individuals who lack the delta 508 change observed in most European individuals. As gene therapy is applied to all F8 pathogenic patients moving forward, balancing population characterization for F8 and phenotypes is critical. Understanding the bleeding and clotting disorders of diverse males and the risks for phenotypic changes in females is needed.

The U.S. hemophiliac population is notably diverse, with significant representation of different ethnic groups [57]. However, this diversity presents challenges in using novel therapies and treatment strategies across the patient population [58]. The study of population pharmacokinetic models of various F8 products developed using diverse populations highlights the variability in dosing required for personalized prophylaxis in hemophilia A [59]. We anticipate that implementing the more expensive one-time administration of gene therapy will further highlight the diversity challenges in F8 amino acid variant analysis and treatments [60].

Historically, females carrying a single copy of F8 with a variant or variants were considered asymptomatic carriers due to a second wild-type F8 protein maintaining function. However, emerging evidence suggests that female carriers exhibit varying degrees of bleeding diathesis [61] and can have non-random XCI that results in a single damaging allele to drive the disease state [14,15,16,17,18]. Female carriers have lower levels of F8 compared to non-carriers [62] and can have diminished F8 response to desmopressin, a standard treatment for hemophilia A [63], possibly contributing to increased bleeding. Furthermore, it has been reported that female carriers of null variants with a BAT score ≥ 6 showed a median factor level of 34 I.U./dL. In contrast, carriers of non-null variants with a BAT score ≥ 6 had a median factor level of 13 I.U./dL [19]. We also highlight in Figure 2D–G that eQTLs present on the functional allele of a carrier could also impact the overall aspects of phenotype. This highlights the substantial impact of F8 variants on factor levels in females, underscoring the need for comprehensive molecular diagnosis in each case, given the historical reliance on factor VIII activity assays for diagnosing hemophilia A and the recognized clinical and therapeutic heterogeneity in females.

The data in Figure 2 and Table 1 show the lack of existing genotype-to-phenotype data within females. While many GWAS have been performed for bleeding phenotypes, the statistical analyses of GWAS require allele frequencies far greater than those in Table 1. Also, few of these GWAS have performed the more complex modeling required for sex chromosome analysis, in which many Chromosome X variants end up masked due to the heterozygous and hemizygous complexities. The analysis shows that few phenotypes have been addressed for rare variants with a gene burden and functional amino acid assessment focused on females with F8 variants. As the phenotypes may be more subtle, there is a need to understand how F8 variants may impact women’s physiologies.

Our work also highlights other challenges in F8 genetics. As Figure 1 and Figure 2 show, the short F8-201 isoform, known as F8B, is highly and ubiquitously expressed with an altered signal peptide. Expression of the F8B in mice can result in significant developmental phenotypic changes [54], suggesting the transcript is not inert. The mouse phenotype was shown over two decades ago, and the protein has remained poorly defined. The growth of epigenetics data suggesting a critical secondary promoter for this short isoform and the ubiquitous expression profile suggest a critical need to define the F8B isoform and how variants may alter its function.

One of the most significant observations from this work highlights the need for ClinVar, ClinGen, or a new tool to continue annotating variant outcomes following deposit. Our work shows that multiple pathogenic and likely pathogenic variants deposited into ClinVar have little support for functionality, while multiple VUS are highly probable for disease. As ClinVar depositing is based on the user, no current organized annotations exist. However, few scientists realize that ClinVar is not definitive in annotation and must be used carefully when assessing variants. The outstanding tools of ClinVar and Geno2MP paired with gnomAD and other population genomics create an incredible resource for exploring variants for a protein. Still, as we show in this paper, much work needs to be performed to logistically organize and define the phenotypic landscape of genetic variants for any protein.

## 5. Conclusions

The data on gene regulation and amino acid variants highlights the two unique isoforms of F8, which are coded by two different promoter elements. The roles of eQTLs and other variants in influencing expression remain poorly understood, particularly for female disease carriers. The analysis of missense variants highlights the lack of phenotyping for females and shows the challenges associated with annotated variants in diverse populations. VUS that are ultra rare also remain an incredible challenge in F8 biology, where tools like molecular dynamics simulations paired with functional prediction algorithms can better define the connection of VUS to highly functional pathogenic F8 variants. Even though F8 is one of the most published and well-explored proteins in the human genome, as our work highlights, there is an incredible need to organize and curate data. As new expensive, one-time treatments such as gene therapy become available, we must advance our data organization and characterization of variants to prioritize the needs of the field.

## Figures and Tables

**Figure 1 genes-15-01522-f001:**
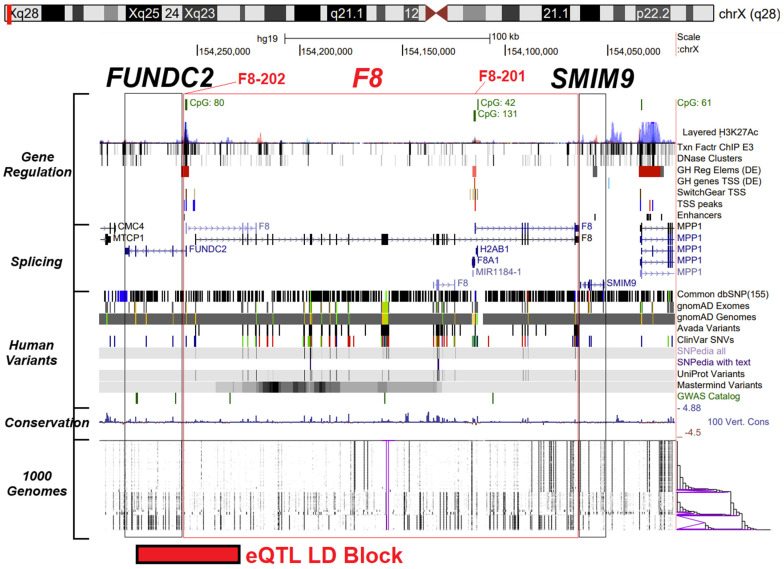
Human genomic data for the *F8* region of Chromosome X, based on the hg19 genome build. The genome browser view of *F8* (red box), flanked by *FUNDC2* and *SMIM9* (black boxes). On the left, the various datasets are marked, including the annotations for gene regulation (H3K27Ac, marking enhancers and transcription-factor binding-site regions; DNase clusters marking regions of open gene regulation, GeneHancer regulatory elements and transcriptional start sites, and SwitchGear transcriptional start sites, transcriptional start site peaks, and known enhancers), splicing sites (exons marked in boxes and arrows indicating transcription direction), various human variant databases, conservation from 100 vertebrate species, and the 1000 genomes variant-linkage maps. The *F8-201* and *F8-202* isoforms are marked at the top of the viewer.

**Figure 2 genes-15-01522-f002:**
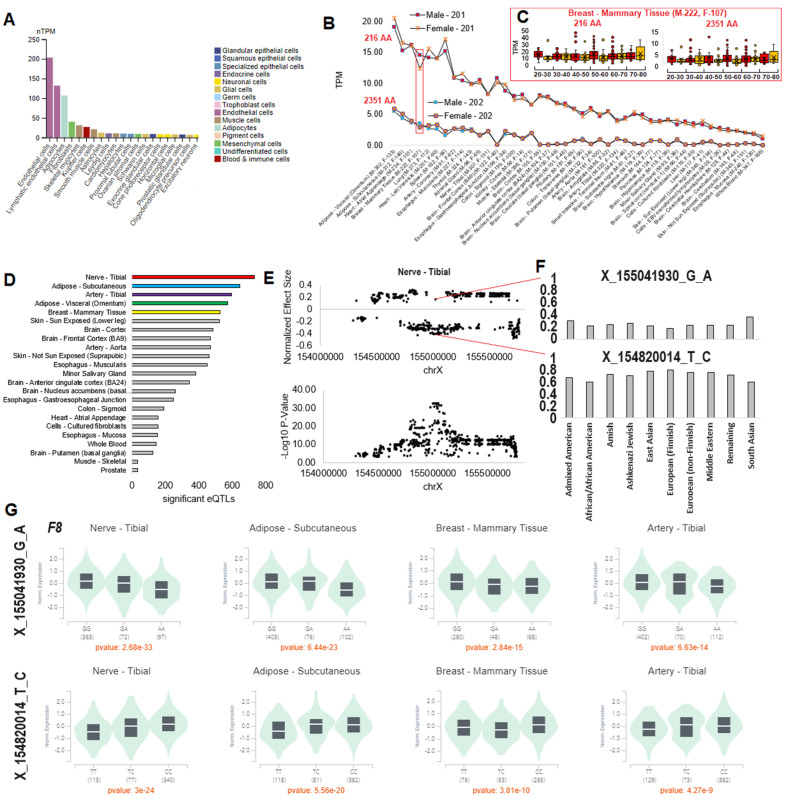
Tissue expression of the *F8-201* and *F8-202* isoforms. (**A**) Top single-cell expression of *F8*, as extracted from the human protein atlas. (**B**) *F8* expression of the 201-isoform coding for a 216 amino acid protein and the 202-isoform coding for a 2351 amino acid protein, in various tissues of the GTEx database. Male samples are shown with boxes associated with their values, and females are shown with an “x”. The red box marks the panel C area of identification. (**C**) Values for male (red) and female (orange) samples at various ages for *F8* expression in breast–mammary tissues, as shown by box-and-whisker plot. (**D**) The number of significant eQTLs in various tissues of GTEx. (**E**) The normalized effect-size (top) and −log10 *p*-values for nerve–tibial eQTLs at various X-chromosome locations. The top two variant eQTLs are identified in panel F. (**F**) The gnomAD-based allele frequencies for the top two eQTLs are shown for various population groups. (**G**) Violin plots for the two variants of panel F in four different tissues, showing the homozygous and heterozygous variant effect on normalized *F8* expression, indicating the potential for influence on *F8* gene regulation.

**Figure 3 genes-15-01522-f003:**
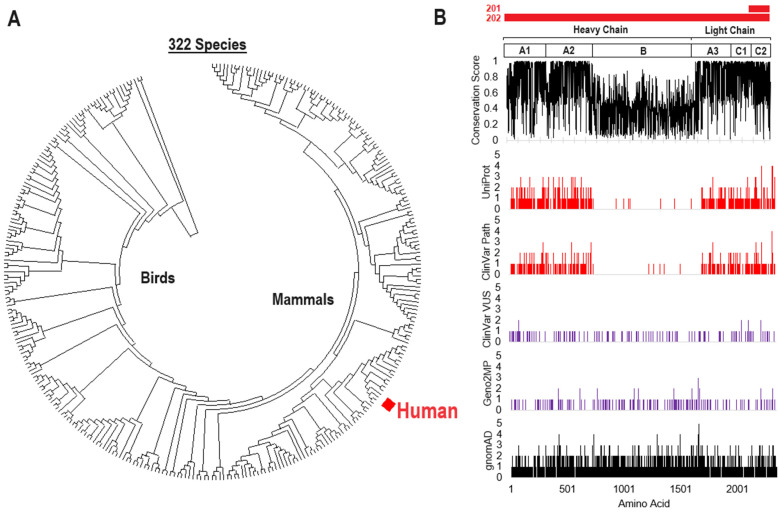
Vertebrate evolution and variant extraction for F8. (**A**) Phylogenic tree of the F8 protein sequences of 322 species, generated by bootstrap consensus. The human F8 protein is marked in red. (**B**) Amino acid and variant locations on the human F8 protein. The red bars at the top show the locations of the 201- and 202-isoform encoded F8 proteins. Domain annotation is shown below, followed by the conservation score, in which 1 represents 100% of species having the same amino acid as that found in humans. The bottom plots show the number of amino acid changes at each amino acid of human F8, with the red plots representing known disease states (from UniProt and ClinVar, pathogenic or likely pathogenic); the magenta, those with potential phenotypes or disease states that are uncertain (from ClinVar as variants of uncertain significance or in Geno2MP); and the black plot, from gnomAD, representing population-level variation.

**Figure 4 genes-15-01522-f004:**
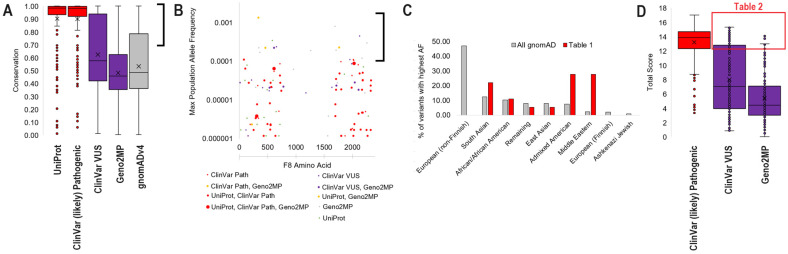
(**A**) The conservation score for each variant from the various databases is shown as a box-and-whisker plot, x indicates the median value. (**B**) The variants in UniProt, ClinVar, or Geno2MP with conservation scores greater than 0.7 (>70% of species conserved to humans) are also found in the gnomAD database. The *x*-axis shows the variant’s location, and the *y*-axis shows the maximum allele frequency observed within one of the populations of gnomAD. (**C**) The population identified in panel B, for those variants with maximum allele frequencies greater than 0.001 (red) relative to all variants seen in gnomAD (gray). The variants in red are listed in Table 1. (**D**) A combined variant score from various functional prediction tools, shown as ClinVar pathogenic or likely pathogenic (red) relative to ClinVar VUS or Geno2MP variants. Variants are shown as a box-and-whisker plot. Those variants with scores similar to pathogenic variants are listed in Table 2.

**Figure 5 genes-15-01522-f005:**
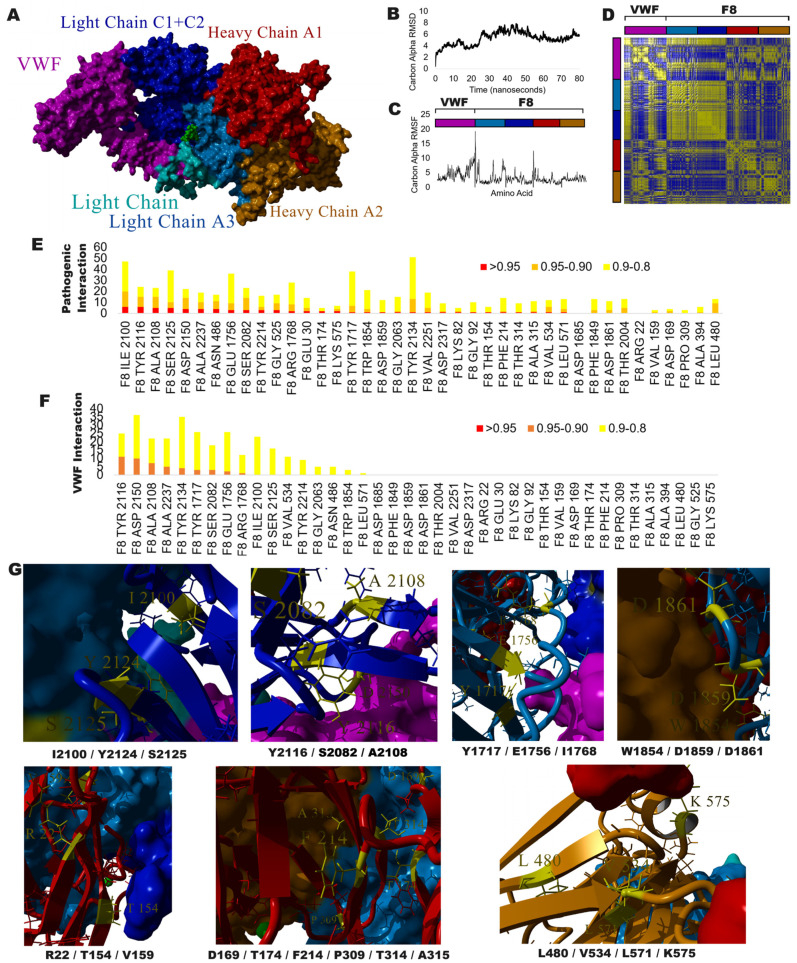
(**A**) Protein model of the von Willebrand factor (VWF, magenta) interacting with the F8 light-chain (cyan), light-chain A3 (light blue), light-chain C1 and C2 (blue), heavy-chain A1 (red), and heavy-chain A2 (orange). (**B**) Global movement of all protein amino acids over an 80-nanosecond molecular dynamics simulation. (**C**) Each amino acid movement over the simulation; the domains are labeled as in panel A. (**D**) Dynamic Cross Correlation matrix of movement for each amino acid with every other amino acid. Sites in yellow have a high correlation, and sites in blue have a low correlation. (**E**) The number of pathogenic and likely pathogenic ClinVar amino acids, with various correlations to VUS of Table 2. (**F**) The number of VWF amino acids with various correlations to VUS of Table 2. (**G**) Structural location of top VUS (yellow) relative to protein domains and pathogenic variants (side chains shown).

**Table 1 genes-15-01522-t001:** F8 missense variants at conserved amino acids found in the gnomAD database.

			gnomAD	gnomAD	gnomAD	gnomAD	gnomAD	Geno2MP	ClinVar and Geno2MP
Variant	Conservation	Group	Homozygote	Hemizygote	Overall AF	Max AF	Population	Homozygous	Phenotypes
E340K	0.96	UniProt, Geno2MP	0	45	7.61E-05	1.39E-03	South Asian	2	Nephrotic syndrome, Abnormal muscle physiology
E1690G	0.87	Geno2MP	0	2	2.37E-05	8.28E-04	East Asian	0	Bladder exstrophy
S2125T	0.96	ClinVar VUS	0	14	3.73E-05	4.58E-04	Middle Eastern	0	Hereditary factor VIII deficiency disease
N583S	0.99	UniProt	1	17	1.82E-05	3.52E-04	South Asian	0	-
T2256S	0.86	Geno2MP	0	3	1.16E-05	3.06E-04	Admixed American	1	Fatigable weakness
Y657H	0.94	ClinVar VUS	0	3	6.38E-06	2.42E-04	Middle Eastern	0	Hereditary factor VIII deficiency disease
Q1764R	0.89	ClinVar Path, Geno2MP	0	56	1.32E-04	2.31E-04	Remaining	1	Thrombophilia, Heterotaxy
R458H	0.93	ClinVar Path, Geno2MP	0	20	4.96E-05	2.29E-04	Middle Eastern	0	Abnormality of hindbrain morphology
R612H	0.91	ClinVar VUS, Geno2MP	0	7	2.48E-05	2.29E-04	Middle Eastern	1	Aplasia cutis congenita, Hereditary factor VIII deficiency disease
K344Q	0.77	Geno2MP	0	4	9.92E-06	2.28E-04	Middle Eastern	0	Ebstein’s anomaly of the tricuspid valve
P1707L	0.82	Geno2MP	0	19	4.88E-05	1.98E-04	Admixed American	3	Retinal degeneration
P1265Q	0.75	Geno2MP	0	6	1.24E-05	1.94E-04	South Asian	0	Nephrotic syndrome
D364N	0.74	Geno2MP	0	7	1.16E-05	1.75E-04	Admixed American	0	Progressive muscle weakness
R2166Q	0.92	Geno2MP	0	15	4.64E-05	1.23E-04	South Asian	2	Abnormality of limbs
P83R	0.83	UniProt, ClinVar Path	0	3	8.22E-06	1.14E-04	Admixed American	0	Hereditary factor VIII deficiency disease
R2016Q	0.95	ClinVar Path	0	3	8.20E-06	1.14E-04	African/African American	0	Hereditary factor VIII deficiency disease
Q1955E	0.92	ClinVar VUS	0	0	2.73E-06	1.14E-04	African/African American	0	Hereditary factor VIII deficiency disease
E2023K	0.95	ClinVar Path	0	4	1.16E-05	1.09E-04	Admixed American	0	Thrombophilia

**Table 2 genes-15-01522-t002:** Highly ranked F8 missense variants of uncertain significance (VUS) in ClinVar, as well as those found within Geno2MP.

Variant	Group	322 Species Conservation	Total Score	gnomAD Allele Count	Geno2MP Homozygous	gnomAD Homozygote	CADD_PHRED	SIFT	PolyPhen
P309L	ClinVar VUS	0.95	15.31	0	0	0	26.2	deleterious	probably_damaging
W1854C	ClinVar VUS	1.00	15.23	0	0	0	26.4	deleterious	probably_damaging
Y2134H	ClinVar VUS	1.00	15.00	0	0	0	26.1	deleterious	probably_damaging
Y2116H	ClinVar VUS	0.99	14.99	0	0	0	26.8	deleterious	probably_damaging
D2150V	ClinVar VUS	0.99	14.92	0	0	0	28.6	deleterious	probably_damaging
D169N	ClinVar VUS	1.00	14.85	0	0	0	25.6	deleterious	probably_damaging
T174I	ClinVar VUS	1.00	14.77	0	0	0	24.7	deleterious	probably_damaging
G92S	ClinVar VUS	1.00	14.76	0	0	0	24.8	deleterious	probably_damaging
A2237S	ClinVar VUS	0.99	14.55	0	0	0	24.9	deleterious	probably_damaging
L571R	ClinVar VUS	1.00	14.50	0	0	0	26.4	deleterious	probably_damaging
T2004A	ClinVar VUS	1.00	14.45	0	0	0	25.7	deleterious	probably_damaging
D1861G	ClinVar VUS	0.87	14.43	0	0	0	26.1	deleterious	probably_damaging
N486I	ClinVar VUS	0.99	14.24	0	0	0	26.1	deleterious	probably_damaging
V2251M	ClinVar VUS	0.98	14.18	0	0	0	24.6	deleterious	probably_damaging
T314P	ClinVar VUS	0.99	14.17	0	0	0	25.2	deleterious	probably_damaging
I2100T	ClinVar VUS	0.89	14.13	0	0	0	24.4	deleterious	probably_damaging
G525R	ClinVar VUS	0.92	14.12	4	0	0	25.1	deleterious	probably_damaging
G2063E	Geno2MP	0.98	14.07	0	0	0	28.8	deleterious	probably_damaging
L480P	ClinVar VUS	0.61	14.06	0	0	0	25.7	deleterious	probably_damaging
E1756V	ClinVar VUS	0.95	13.97	0	0	0	27.4	deleterious	probably_damaging
S2125T	ClinVar VUS	0.96	13.94	45	0	0	24.7	deleterious	probably_damaging
A2108P	ClinVar VUS	0.99	13.85	0	0	0	24.2	deleterious	probably_damaging
R1768S	Geno2MP	0.98	13.71	0	0	0	25.6	deleterious	probably_damaging
Y1717H	ClinVar VUS, Geno2MP	1.00	13.65	3	0	0	25.8	deleterious	probably_damaging
R22K	ClinVar VUS	0.98	13.43	0	0	0	25.9	deleterious	probably_damaging
F1849I	ClinVar VUS	0.98	13.43	0	0	0	26	deleterious	probably_damaging
E30G	ClinVar VUS	0.96	13.35	0	0	0	23.7	deleterious	probably_damaging
K82E	ClinVar VUS	0.98	13.28	0	0	0	24.8	deleterious	probably_damaging
A394D	ClinVar VUS	0.97	13.27	0	0	0	25.2	deleterious	probably_damaging
D1859V	ClinVar VUS	0.58	13.24	0	0	0	26	deleterious	probably_damaging
K575T	ClinVar VUS	0.98	13.13	1	0	0	24.6	deleterious	probably_damaging
T154I	ClinVar VUS	0.96	13.05	1	0	0	24.3	deleterious	probably_damaging
S2082N	ClinVar VUS, UniProt	0.97	13.03	0	0	0	24.6	deleterious	probably_damaging
A315G	ClinVar VUS	0.98	13.01	0	0	0	24.6	deleterious	probably_damaging
Y2214C	Geno2MP	0.70	12.98	7	0	0	25.6	deleterious	probably_damaging
D1685G	Geno2MP	0.70	12.86	0	2	0	25.2	deleterious	probably_damaging
V534A	ClinVar VUS, Geno2MP	0.74	12.80	2	1	0	25.6	deleterious	probably_damaging
A2108V	ClinVar VUS, Geno2MP	0.99	12.56	1	1	0	25.4	deleterious	possibly_damaging
D2317A	Geno2MP	0.62	12.50	3	1	0	25.2	deleterious	probably_damaging
F214Y	ClinVar VUS	0.92	12.49	1	0	0	24.9	deleterious	probably_damaging
V159A	ClinVar VUS, UniProt	0.52	12.33	0	0	0	24.9	deleterious	possibly_damaging

## Data Availability

Raw organized data are available at https://doi.org/10.6084/m9.figshare.26499532.v1.

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
