# Peer review of "Genomic Landscape of Chromosome X Factor VIII: From Hemophilia A in Males to Risk Variants in Females"

_genes, 2024, doi:10.3390/genes15121522_

Round 1

Reviewer 1 Report

Comments and Suggestions for Authors

In this study the authors analysed F8 variants considering their structure and frequency in different populations. One important result, obtained by the authors, is the possible functional relevance of some ultrarare F8 variants annotated as VUS in the ClinVar database. In addition, the authors criticize the classification of some F8 variants annotated as pathogenic in ClinVar.

The paper is interesting for readers that want to know more about the classification of F8 genetic variants. In particular, protein-based molecular dynamic simulations made by the authors can give additional information for the interpretation and classification of F8 variants.

I have just some comments that the authors could be considered to further improve the discussion.

The authors correctly underline that not every classification in ClinVar should be considered as correct, but it should be noted that ClinVar has a system to define the quality of the classifications. In fact, the five examples reported by the authors with a possible wrong classification are all with one star (classification made by only one laboratory). In two of them (Q176R and E2023K) the status of the sequenced subject is unknown but in the other three (P83R, R458H, R2016H) the sequence is from an affected subject. This should be taken in consideration because some pathogenic variants could be at low penetrance.

It is not surprising that VUS can be upgraded to like path/path variants because some laboratories prefer a cautious approach in the classification without strong functional and clinical data. About 10% of VUS will be upgraded to like path/path variants (PMID: 31752965); the authors could estimate and discuss the percentage of F8 VUS that they think could reach the pathogenic tier.  

Author Response

The authors correctly underline that not every classification in ClinVar should be considered as correct, but it should be noted that ClinVar has a system to define the quality of the classifications. In fact, the five examples reported by the authors with a possible wrong classification are all with one star (classification made by only one laboratory). In two of them (Q176R and E2023K) the status of the sequenced subject is unknown but in the other three (P83R, R458H, R2016H) the sequence is from an affected subject. This should be taken in consideration because some pathogenic variants could be at low penetrance.

Thank you for your insightful feedback. We agree that declassification should be taken on carefully due to penetrance rates and want to be clear that we are not definitively arguing for a change in classification. We have reworded to say: "The data for Q1764R, R458H, R2016Q, and E2023K suggests further examination of their annotation in ClinVar may be warranted."  line 292-293

It is not surprising that VUS can be upgraded to like path/path variants because some laboratories prefer a cautious approach in the classification without strong functional and clinical data. About 10% of VUS will be upgraded to like path/path variants (PMID: 31752965); the authors could estimate and discuss the percentage of F8 VUS that they think could reach the pathogenic tier.  

We also wholeheartedly agree with your second comment regarding upgraded classification, and a cautious approach. It would be a great addition to the paper to estimate and discuss the percentage of F8 VUS that could reach path classification but is out of the scope of this paper. We will definitely keep in mind for future work. Thank you again for taking the time to provide valuable feedback, we know you have a busy schedule.

Reviewer 2 Report

Comments and Suggestions for Authors

This paper has a thorough, multi-angled approach to amino acid function and conservation, and highlights a specific need of annotation in non-European (ancestry) individuals and datasets. The paper has also an excellent supplementary materials breakdown.

Just a few observations:

62: while this is technically correct, perhaps you should add “deleterious” to properly define the compound. Otherwise, most individual could be technically considered CH for a large number of genes.

As an additional note – perhaps it could be explained in the discussion, but these isoforms are majorly different in size. Could you elaborate on the whys?  

149: here, and in the rest of the manuscript – I think you should use accession numbers for your isoforms. Are these ENSEMBL ones? What is the NP for the proteins?

176 and 177 – minor, but that’s a lot of “however”

182: this type of statements belongs more to the discussion

Figure 2 G – the effect is far from being clear. Is there space to elaborate about that in the figure caption, even briefly?

Figure 3 (and text) – I think the vertebrate evolution analysis is a great idea but is undercooked. It should be better integrated into

433: consider on whether “mutated” here is accurate.

Author Response

62: while this is technically correct, perhaps you should add “deleterious” to properly define the compound. Otherwise, most individual could be technically considered CH for a large number of genes.

Thank you for providing us with encouragement and helpful feedback.   We agree with your comment on the technicality of adding deleterious to more fully describe the compound and have made this change in the paper.   In some sporadic cases, homozygous, compound heterozygous (meaning two separate deleterious variants on opposite ChrX), female hemizygous (lacks a second copy of the gene), or heterozygous variants with non-random XCI occur in females with Hemophilia A [14–18].   line 62

As an additional note – perhaps it could be explained in the discussion, but these isoforms are majorly different in size. Could you elaborate on the whys?  

We felt that your questioning on the size of the isoforms was spot on and have elaborated on the difference in size in the body of the paper. Thank you for this helpful suggestion.  

F8-202 translation results in a 2,351 amino acid protein, while the F8-201 isoform codes for a 216 amino acid protein. F8-202 encodes a large glycoprotein which associates with von Willebrand factor in a noncovalent complex. F8-201 encodes a putative small protein mainly consisting of the phospholipid binding domain of factor VIIIc and is essential for coagulant activity.      line 155-157

149: here, and in the rest of the manuscript – I think you should use accession numbers for your isoforms. Are these ENSEMBL ones? What is the NP for the proteins?

  We agree that adding the accessions numbers for the 2 isoforms is needed and have included them in the first reference to each.   F8 has two major promoter elements, one located near the FUNDC2 gene that codes for the F8-202 isoform ( NP_000123) and another promoter internal to the gene that codes for the F8-201 isoform(NP_063916).     line 148-149

176 and 177 – minor, but that’s a lot of “however”

We have addressed the excess number of howevers in lines 176 and 177. Thank you! Much better read.

However, in the breast–mammary tissue, there is a trend for higher expression of both isoforms in males. Yet, when ...

182: this type of statements belongs more to the discussion

We agree that the statement in line 182 leans more toward discussion than results, and have adjusted the statement to concern results only.   As female samples rarely have double the expression of male samples, it suggests that F8 XCI is consistent across age groups and tissues. Data shows males tend to have more outliers of both isoforms over various age groups.

Figure 2 G – the effect is far from being clear. Is there space to elaborate about that in the figure caption, even briefly?

We agree that the effect from Figure 2 G needed more elaboration and have added to the caption as follows:   G) Violin plots for the two variants of panel F in four different tissues of the homozygous and heterozygous variant effect on normalized F8 expression indicate potential for influence on F8 gene regulation.

Figure 3 (and text) – I think the vertebrate evolution analysis is a great idea but is undercooked. It should be better integrated into

Thank you for your encouragement about the vertebrate evolution analysis. We agree that this very interesting section of the paper should be expanded

433: consider on whether “mutated” here is accurate.

We agree that the term "mutated" should be removed from line 433 and have reworded as: Historically, females carrying a single copy of F8 with a variant or variants were considered asymptomatic carriers due to a second wild-type F8 protein maintaining function.